# Plant-Derived UDP-Glycosyltransferases for Glycosylation-Mediated Detoxification of Deoxynivalenol: Enzyme Discovery, Characterization, and In Vivo Resistance Assessment

**DOI:** 10.3390/toxins17040153

**Published:** 2025-03-22

**Authors:** Valeria Della Gala, Laura Dato, Gerlinde Wiesenberger, Diana Jæger, Gerhard Adam, Jørgen Hansen, Ditte Hededam Welner

**Affiliations:** 1Novo Nordisk Foundation Center for Biosustainability, Technical University of Denmark, Søltofts Plads 220, DK-2800 Kongens Lyngby, Denmark; vdellag@biosustain.dtu.dk; 2River Stone Biotech ApS, Fruebjergvej 3, DK-2100 Copenhagen, Denmark; laurad@gly-it.com (L.D.); dianaj@rstbio.com (D.J.); jorgenh@rstbio.com (J.H.); 3Institute of Microbial Genetics, Department of Agricultural Sciences, BOKU University, Konrad Lorenz Strasse 24, AT-3430 Tulln, Austria; gerlinde.wiesenberger@boku.ac.at (G.W.); gerhard.adam@boku.ac.at (G.A.)

**Keywords:** UDP-glycosyltransferases, UGTs, glycosylation, plant, *Fusarium*, mycotoxin, deoxynivalenol, DON, detoxification

## Abstract

Fungal infections of crops pose a threat to global agriculture. Fungi of the genus *Fusarium* cause widespread diseases in cereal crops. *Fusarium graminearum* reduces yields and produces harmful mycotoxins such as deoxynivalenol (DON). Plants mitigate DON toxicity through glucose conjugation mediated by UDP-glycosyltransferases (UGTs), forming deoxynivalenol-3-*O*-glucoside (DON-3-Glc). Few such UGTs have been identified, predominantly from *Fusarium*-susceptible crops. Given that the presence of this activity in diverse plants and across broader UGT subfamilies and groups was underexplored, we screened a library of 380 recombinant plant UGTs and identified and characterized eight novel enzymes glycosylating DON in vitro. Among these, *Zj*UGT from *Ziziphus jujuba* stood out with the highest activity, showing an apparent *k*_cat_ of 0.93 s^−1^ and *k*_cat_/*K*_m_ of 2450 M^−1^ s^−1^. Interestingly, four enzymes produced primarily a novel, still uncharacterized glucoside. Furthermore, we evaluated the in vivo resistance provided by these UGTs when expressed in a DON-sensitive yeast strain. At least six of the novel UGTs conferred some level of resistance, allowing growth at concentrations of up to 120 mg/L of DON. This study contributes to potential strategies to enhance DON resistance in cereal crops in the future.

## 1. Introduction

The global human population is expected to grow from the current 8 billion to 9.7 billion by 2050, posing challenges to food security and sustainability [1]. According to the Food and Agriculture Organization (FAO), food production must increase by 60% to meet these demands [2]. However, crop infections, exacerbated by climate change, pose a significant threat to crop production, with fungal infections causing yield losses of 10–20% annually, further worsened by post-harvest losses [3,4,5,6,7]. Rice, maize, wheat, and barley are among the most commonly cultivated and consumed primary crops across the globe [8]. These crops are grown for both human and animal consumption, and they are susceptible to *Fusarium graminearum*, a fungus that causes Fusarium head blight (FHB) and Fusarium crown rot (FCR) diseases, leading to significant losses in grain yield and reduced crop quality [9,10]. In addition to crop losses, the production of mycotoxins by these fungi poses a significant threat to food and feed safety, with trichothecenes, particularly deoxynivalenol (DON), being among the most harmful to humans and animals [11,12]. DON is a virulence factor of *Fusarium* suppressing plant defense [13]. At the cellular level, DON is a potent ribotoxin that disrupts protein synthesis, posing significant health risks [14,15,16,17,18]. DON primarily targets the gastrointestinal system, impairing nutrient absorption and weakening intestinal barriers [19]. The European Union has set mycotoxin limits in food and feed, respectively (Commission Regulation (EC) Nos. 1126/2007 and 576/2006), but these thresholds are often exceeded [20,21]. Thus, understanding the mechanisms that plants have evolved to resist fungal infections and mycotoxin accumulation is essential for developing innovative crop protection strategies. Specifically, one effective detoxification strategy to mitigate the harmful effects of mycotoxins involves their conjugation with sugars, which reduces their toxicity and bioavailability [22]. This process, known as phase II detoxification, is catalyzed by uridine diphosphate (UDP)-dependent glycosyltransferases (UGTs). UGTs are ubiquitous enzymes belonging to the glycosyltransferase (GT) Family 1 within the CAZy database (www.cazy.org) [23,24], and they transfer sugar moieties from nucleotide sugars to diverse acceptors, catalyzing the formation of *O*-, *N*-, *S*-, and *C*-glycosidic bonds [25,26]. UGTs are structurally characterized by the GT-B fold, comprising two Rossmann-like domains separated by a catalytic cleft [25]. The N-terminal domain interacts primarily with the acceptor substrate, while the C-terminal domain recognizes the sugar donor through a conserved motif named “plant secondary product glycosyltransferase” (PSPG) [27,28]. The catalytic activity relies on a histidine–aspartate dyad, which facilitates nucleophile deprotonation and an S_N_2-like mechanism for sugar transfer [29,30]. UGTs are crucial for mycotoxin detoxification and have been found in many plant species, including *Arabidopsis*, wheat, and rice, where they help mitigate the harmful effects of DON by catalyzing the formation of DON-3-*O*-glucoside (DON-3-Glc, Figure 1) [31,32,33,34,35,36,37]. *Arabidopsis* DOGT1 (*At*UGT73C5) and barley *Hv*UGT13248 were among the first UGTs reported for DON detoxification [31,32], with homologs in *Brachypodium* and wheat also linked to DON tolerance [33,36,38]. Rice *Os*UGT79 was the first UGT to be biochemically characterized for its activity against DON in vitro [34,39,40]. It was also mutationally analyzed to investigate the structural determinants underlying its detoxification potential [35]. Recent studies additionally identified oat UGTs contributing to DON resistance [37]. Notably, *Hv*UGT13248 from barley was shown to confer resistance to both DON and the structurally related mycotoxin nivalenol (NIV) when expressed in wheat, decreasing the severity of both FHB and FCR [41,42]. In barley, functional mutants of *Hv*UGT13248 showed greater susceptibility to FHB and reduced levels of DON-3-Glc, whereas overexpression of the wild-type enzyme improved resistance and increased DON-3-Glc production [43]. Nevertheless, research to date on plant UGTs active on DON, as summarized in Appendix A, remains largely confined to a narrow set of enzymes, primarily from plant species naturally exposed to, or phylogenetically related to, DON-exposed plants [34,39,40,42]. In this study, we employed an unbiased screening approach using a diverse library of promiscuous UGTs that are well-expressed in *Escherichia coli*, expanding the representation of UGT subfamilies and potentially exploring functional diversity beyond the previously characterized groups. Screening this commercial library (GLY-it library) led to the identification of nine UGTs capable of glycosylating DON, eight of which were not previously reported in the literature. These enzymes produced DON-3-Glc and, interestingly, a second glycosylation product of unknown structure (DON-X-G), suggesting an alternative detoxification pathway. Furthermore, their ability to confer resistance in a DON-sensitive yeast strain demonstrated functional relevance beyond in vitro activity. These findings broaden our understanding of UGT-mediated mycotoxin detoxification and highlight promising candidates for further characterization and application in crop protection.

## 2. Results and Discussion

### 2.1. Screening of the GLY-it Library

To identify novel UGTs with the capacity to glycosylate DON, we screened a plant UGT library (GLY-it library) consisting of 380 purified enzymes spanning diverse phylogenetic groups [44]. This commercial library is selected to include well-expressed and substrate-promiscuous enzymes, reflecting the broad catalytic potential of UGTs. Using LC–MS/MS (for conditions, see Appendix A), the disappearance of DON after 24 h was measured, along with the formation of products that had the mass of a DON-glucoside (Appendix A). Interestingly, our LC–MS/MS analysis revealed that DON-3-Glc eluted later than DON, which we attribute to the composition of the mobile phase used in the chromatographic separation. Nine UGTs originating from different plants with activity toward DON were identified (Table 1, Appendix A). These include *At*UGT73C5, a well-characterized enzyme previously reported to detoxify DON [31]. While its presence validated the effectiveness of the screening method and served as a positive benchmark, we focused subsequent characterization efforts on the eight newly identified enzymes that were not previously reported in the literature. While these enzymes have not previously been described to glycosylate DON, some originate from plant species that can be infected by *Fusarium*, such as sesame [45,46,47], red date [48,49], and sugar beet [50,51,52,53]. Sequence analysis showed that these enzymes have an average identity of ~30% to known DON-detoxifying UGTs (Appendix A). The novel UGTs predominantly belong to UGT Subfamily 73, except for *Eg*UGT and *Zj*UGT (Table 1). Amino-acid-based phylogenetic analysis further revealed that most of the newly identified UGTs are only distantly related to previously reported enzymes, underscoring the unique diversity uncovered by the screening (Appendix A). Notably, *Zj*UGT, a member of Subfamily 71, Group E, converted the most DON after 24 h among the nine hits (Appendix A), despite the fact it had an average sequence identity of only ~22% to known DON-detoxifying UGTs (Appendix A).

### 2.2. Characterization of the Novel DON UGTs

The eight UGTs identified in the initial screening were expressed and purified (Appendix A). A time-course activity assay confirmed activity towards DON and production of DON-3-Glc (Figure 2), although *Si*UGT, *Pt*UGT, *Bv*UGT, and *Ac*UGT at low conversion rates (Appendix A). To ensure the reliability of our activity assays, we included controls demonstrating no spontaneous DON degradation or buffer interference, as shown in Appendix A. The best-performing DON-3-Glc producer, *Zj*UGT, converted over 90% of the supplemented 500 μM DON to the glucoside within 24 h, even under non-optimized reaction conditions. Interestingly, the four low producers (*Si*UGT, *Pt*UGT, *Bv*UGT, *Ac*UGT) did not produce DON-3-Glc in equimolar amounts compared to DON consumption, although DON-3-Glc was formed in a time-dependent manner (Appendix A). Instead, these four UGTs converted DON to an unknown product, consistent with observations made during the initial library screening (Appendix A). In addition to DON-3-Glc, two other glucosides are theoretically possible: DON-7-Glc, which, to our knowledge, has not been reported and may be less favorable due to the lower reactivity of the C7–OH, and DON-15-Glc, which has been described in the literature [55,56]. However, our HPLC analysis indicated that DON-X-G had a different retention time than the DON-15-Glc analytical standard (Appendix A). Preliminary QTOF–MS analysis confirmed the presence of a glucose fragment but could not conclusively identify the glycosylation position. Similarly, NMR analyses were not successful in identifying the compound’s identity, due to the amounts available being too low. Despite detectable bioconversion by *Si*UGT and other enzymes, the overall yield of DON-X-G remained insufficient for large-scale purification and high-resolution spectroscopic characterization. Future work involving upscaled enzyme reactions and optimized purification strategies will be required to elucidate the precise structure of this novel detoxification product.

Next, the pH and temperature profiles, along with the thermostability of the top DON-3-Glc-producing UGTs, were systematically evaluated using DON as the acceptor substrate and UDP-Glc as the donor (Table 2). The enzymes demonstrated a consistent pH preference between pH 8 and 8.5 (Appendix A) and had temperature optima between 30 °C and 45 °C, with *Sr*UGT and *Ac*UGT being the more thermophilic enzymes and *Eg*UGT the more psychrophilic enzyme (Appendix A). Typically, UGTs are characterized by limited temperature stability [57], but we found great variability in this respect (Appendix A). *Ac*UGT demonstrated exceptional thermostability, being able to withstand temperatures up to 63.0 °C. Other UGTs exhibited moderate stability, with T_M_ values ranging from 48.1 °C to 59.4 °C. In contrast, *Eg*UGT was the least stable, with a T_M_ value of 46.9 °C, which aligns well with its psychrophilic properties, a phenomenon consistent with trends observed in the literature [58]. The kinetic properties of the top DON-3-Glc producers were also investigated (Appendix A). Most of the UGTs exhibited apparent kinetic parameters consistent with those reported in the literature [34,39,40]. In particular, *Zj*UGT showed the highest *k*_cat_ and catalytic efficiency, and *Ac*UGT the lowest, mirroring the trend of DON-3-Glc formation observed in Figure 2. Among the UGTs known in the literature, only *Os*UGT79 and *Hv*UGT13248 have been kinetically characterized (Table 2). Compared to these, *Zj*UGT exhibited a similar turnover number, although there is some variability in the reported values for *Os*UGT79 (0.57 [34] and 1.07 [40] s^−1^). Assuming that *Hv*UGT13248 was characterized under conditions comparable to those used for *Os*UGT79 [34], *Zj*UGT demonstrates kinetic parameters that are at least equivalent, if not superior, to both enzymes. However, the *K*_m_ values observed for all the identified UGTs were relatively high, similar to other UGT acceptor pairs [59,60,61]. This could be due to the structural or functional constraints of the enzymes, which may have evolved to accommodate a broad range of substrates, thereby enhancing their promiscuity while diminishing their specificity for DON, assumed not to be their natural substrate. However, an enzyme exhibiting a *K*_m_ substantially higher than the concentration at which eukaryotic ribosomes are predominantly inhibited by the toxin (10–100 µM) [62] may fail to confer significant resistance, particularly considering that these UGTs are typically inducible by substrates rather than being constitutively expressed at high levels [63]. Enzyme engineering to optimize substrate recognition and binding could, therefore, prove useful for enhancing plant resistance to DON. Previous efforts to enhance UGT performance through mutagenesis and structure-guided modifications have successfully altered DON specificity and increased catalytic efficiency, providing a promising strategy to overcome the kinetic limitations observed in these enzymes [35].

### 2.3. UGT-Mediated Resistance to DON in Yeast

To functionally evaluate whether these UGTs have the potential to confer resistance to DON beyond in vitro activity, we heterologously expressed both DON-3-Glc- and DON-X-G-producing enzymes in the toxin-sensitive *Saccharomyces cerevisiae* strain YZGA515 [31], a relevant eukaryotic model for cellular detoxification assessment of mycotoxins. The YZGA515 strain is particularly sensitive to DON due to the deletion of the yeast acetyltransferase gene *AYT1*, which prevents the acetylation of DON to 3-acetyldeoxynivalenol, and the inactivation of ABC transporters (*PDR5*, *PDR10*, and *PDR15*), which cause ATP-dependent drug efflux and reduce the toxin concentration on the ribosomal target [64,65]. The transformants expressing the UGTs under a strong constitutive promoter (*ADH1*) were tested for growth in the presence of increasing concentrations of DON (Figure 3). In initial trials, relatively low DON concentrations were used (25–40 ppm, Figure 3A). After three days, some growth was noticed, and after six days, all strains, except for those harboring *Eg*UGT and *Pt*UGT, exhibited growth, even at the highest concentration of DON. The DON-resistant strains from the initial trial were then exposed to higher DON concentrations (40–120 ppm, Figure 3B). After six days, all strains showed growth at 80 ppm DON, with the strains expressing *Si*UGT and *Zj*UGT growing well, almost as well as the strain containing *Hv*UGT13248 as a positive control [32]. This observation is particularly noteworthy given that *Si*UGT primarily generates the still unidentified glucoside (DON-X-G). This indicates that glycosylation of DON at any side group renders it less toxic for yeast, warranting further investigation due to the potential implications for mycotoxin detoxification.

Interestingly, in vitro catalytic efficiency did not always correlate with in vivo resistance. While *Eg*UGT and *Pt*UGT exhibited poor conversion rates in vitro and failed to provide resistance in yeast, *Sr*UGT and *Sp*UGT displayed rapid turnover of DON in vitro yet conferred little resistance after exposure to 80 ppm DON. This apparent decoupling of in vitro activity from in vivo functionality suggests that cellular detoxification efficacy is not solely determined by the inherent catalytic properties of the enzyme itself. These discrepancies highlight the complex challenges in directly applying enzyme kinetics derived from simplified in vitro biochemical assays to predict enzyme behavior within a complex and dynamic cellular environment.

Several potential factors associated with heterologous expression in yeast could contribute to these inconsistencies. While codon optimization is a well-established strategy for improving recombinant protein expression [66], the lack of optimization in these UGTs may only partially account for the observed effects. Beyond translational efficiency, factors such as protein folding, interactions with yeast-specific chaperones, and susceptibility to proteolytic degradation in the yeast endoplasmic reticulum or cytosol could significantly impact enzyme stability and activity [67,68,69]. Additionally, post-translational modifications, particularly glycosylation patterns in yeast that may differ from those in plant cells, could alter UGTs’ conformation, stability, or substrate binding properties, ultimately affecting their in vivo catalytic performance [70,71].

Another major consideration is the fundamental difference between the controlled conditions of in vitro assays and the intricate intracellular environment [72]. In vitro assays are typically performed in defined buffer systems, often under conditions that maximize catalytic rates [73]. However, inside a living yeast cell, UGTs operate within a competitive substrate landscape, where multiple endogenous metabolites may serve as alternative enzymatic targets. If *Sr*UGT and *Sp*UGT possess broader substrate specificity, they may preferentially glycosylate other cellular metabolites over DON, leading to reduced detoxification efficiency. Moreover, the activity of these UGTs within the yeast cell could be further modulated by allosteric regulators [71,74], such as intracellular metabolites, or through feedback mechanisms [75], for example by DON-glucosides themselves. Such regulatory mechanisms would likely be absent and therefore undetectable in standard in vitro assays.

Finally, the availability of essential cofactors within the yeast cell is another important factor influencing in vivo enzyme activity. UGTs require UDP-glucose as a glycosyl donor, and the intracellular availability of UDP-glucose, as well as its compartmentalization within yeast cells, could impact glycosylation efficiency [76]. If UDP-glucose pools are limited, UGTs with high catalytic efficiency might rapidly deplete the available cofactor supply, leading to a bottleneck in glycosylation. Additionally, enzyme localization within the cell relative to the site of DON entry and action could impose spatial constraints; UGTs function predominantly in the cytosol [77], but their effectiveness depends on the intracellular distribution of DON and whether detoxification occurs before significant toxic effects manifest.

Despite variability between in vitro and in vivo results, this study provides a qualitative assessment of UGT-mediated DON detoxification and demonstrates that several enzymes retain activity in an intracellular environment. Importantly, yeast-based screening has proven to be an effective tool for distinguishing UGTs that function only in vitro from those with the potential for in planta detoxification [41,42,43,78,79]. These findings lay the groundwork for future plant studies, where the most promising UGT candidates should be evaluated for their ability to enhance *Fusarium* resistance in crops. Moreover, the future identification of DON-X-G raises the possibility of an alternative glycosylation-based detoxification mechanism, warranting further structural and functional characterization.

## 3. Conclusions

This study expanded the known repertoire of DON-glycosylating UGTs by identifying and characterizing eight novel enzymes, providing new insights into their catalytic properties and potential applications. Among them, *Zj*UGT, a member of UGT Subfamily 71, Group E, was the most efficient, with a catalytic efficiency of 2450 M^−1^ s^−1^, comparable to the well-characterized *Os*UGT79. Phylogenetic analysis clustered most UGTs within Subfamily 73, aligning with known DON-detoxifying enzymes. In addition, the identification of a previously unreported DON glucoside (DON-X-G) raised new questions regarding alternative detoxification mechanisms, warranting further structural characterization. Functional assessment in a mycotoxin-sensitive yeast strain revealed that most UGTs provided some level of resistance to DON, demonstrating their detoxification potential beyond in vitro activity. These findings greatly expand our understanding of DON-glycosylating UGTs, shedding light on underexplored enzyme subfamilies and groups and providing a foundation for improving DON resistance in crops. Future studies should explore how in vitro activity and yeast resistance translate to in planta detoxification, aiming to harness UGTs for agricultural applications.

## 4. Materials and Methods

### 4.1. Reagents and Chemicals

Unless otherwise mentioned, buffers, standard reagents, DON, and DON-3-Glc were purchased from Sigma-Aldrich (Saint Louis, MO, USA). DON-15-Glc was purchased from Pribolab (Qingdao, China). For the yeast spotting experiments, DON was purchased from TripleBond Canada (Guelph, ON, Canada). GLY-it plasmids carrying UGT sequences with an N-terminal His-tag were provided by River Stone Biotech ApS (Copenhagen, Denmark).

### 4.2. Preliminary Screening of the GLY-it Library

A large commercial UGT library (GLY-it library) containing 380 purified plant enzymes was tested in 96-well plates through an enzymatic reaction with DON in a single measurement. In brief, 20 μL reactions were prepared with 0.5 mM DON and 1.25 mM UDP-Glc in a 100 mM Tris-HCl, 5 mM MgCl_2_, and 1 mM KCl buffer, pH 7.4. Additionally, the buffer contained 0.2 U FastAP thermosensitive alkaline phosphatase (Thermo Fischer Scientific, Waltham, MA, USA) to prevent any inhibition of glycosyltransferase activity. Controls included all the reaction components, including the buffer in which enzymes are resuspended, except for the enzymes. Reactions were initiated by adding 5 μL enzyme, incubated at 30 °C and quenched after 24 h with 60 μL of 75% EtOH.

### 4.3. In Silico Sequence Analysis

The selected sequences were subjected to multiple sequence alignment (MSA) and phylogenetic analysis. MSA and the corresponding identity matrix were constructed using Clustal O, employing the neighbor-joining method [80]. Visualization of the identity matrix and the phylogenetic tree was performed in Rstudio (Rstudio version 2024.04.2 +764, RStudio, Inc., Vienna, Austria) and iTOL, respectively, utilizing default settings [81,82].

### 4.4. Protein Expression in E. coli and Purification

Positive hits from the GLY-it library were expressed and purified in our laboratory for further characterization. In brief, One Shot™ BL21 Star™ (DE3) chemically competent *E. coli* cells (Thermo Fisher Scientific, Waltham, MA, USA) were transformed with GLY-it plasmids. Transformed cells were selected on LB plates supplemented with 50 μg/mL ampicillin and grown overnight at 37 °C in LB medium supplemented with 50 μg/mL ampicillin. Main cultures were prepared by adding 5 mL overnight culture to 500 mL of 2xYT medium with 50 μg/mL ampicillin and grown at 37 °C and 220 rpm. When the culture reached OD_600_ = 0.5–0.7, expression was induced with 0.5 mM IPTG and grown for about 22 h at 20 °C. The cells were harvested by centrifugation at 4500× *g* at 4 °C and resuspended in a buffer containing 50 mM sodium phosphate, 300 mM NaCl, pH 7.5, with 20 mM imidazole supplemented with 0.4 mg DNaseI. The cell suspensions were subjected to cell disruption by sonication (85% nominal power, 30 s ON/30 s OFF for a total of 10 min) on ice using Sonics VCX-130 Ultrasonic Processor (Medline Scientific, Chalgrove, UK). The soluble cell lysate was recovered by centrifugation at 14,500× *g* at 4 °C for 50 min. The supernatant containing the recombinant proteins was filtered with a 0.45 μM syringe filter and purified by nickel-affinity chromatography with a 1 mL prepacked HisTrap^TM^ FF column (Cytiva, Marlborough, MA, USA) on an ÄKTA pure chromatography system (Cytiva, Marlborough, MA, USA). Protein elution was achieved using an imidazole gradient increasing from 20 to 500 mM. The eluted proteins were concentrated using Amicon^®^ Ultra-15 centrifugal filters (30 kDa cutoff), buffer-exchanged to a buffer containing 25 mM sodium phosphate, 150 mM NaCl, pH 7, and their concentration was measured by spectrophotometric measurements at 280 nm. Purity was assessed by SDS-PAGE.

### 4.5. Time-Course Activity Assay with DON

Purified protein samples were used in a total reaction volume of 10 μL to perform in vitro glycosylation reactions. The reaction mixture contained 0.75 mg/mL of enzyme, 0.5 mM DON, and 2 mM UDP-Glc in a 50 mM sodium phosphate buffer, pH 7.5. The control was made with the reaction mixture without the enzyme. The reactions were incubated at 22 °C and quenched at various time intervals (0.5, 1, 3, 6, and 24 h) by adding 90 μL 100% MeOH.

### 4.6. Biochemical and Kinetic Characterization

For the identification of the pH optimum, enzyme activity was investigated in 50 mM sodium citrate (pH 4, 5, 6), sodium phosphate (pH 6.5, 7), HEPES (pH 7.5, 8, 8.5), and glycine (pH 9, 9.5, 10) buffers. Each reaction contained 0.15 mg/mL enzyme, 0.25 mM DON, and 1 mM UDP-Glc in a final volume of 10 μL. The reactions were incubated at 22 °C and quenched after 1 h by adding 90 μL 100% MeOH.

For the identification of the temperature optimum, enzyme activity was investigated in a 50 mM sodium phosphate buffer, pH 7.5. Each reaction contained 0.15 mg/mL enzyme, 0.25 mM DON, and 1 mM UDP-Glc in a final volume of 10 μL. The reactions were carried out at 20–54 °C in a thermocycler and quenched after 1 h by adding 90 μL 100% MeOH.

To determine the melting temperature, differential scanning fluorimetry (DSF) was performed using the Protein Thermal Shift Dye Kit (Thermo Fisher Scientific, Waltham, MA, USA) with a QuantStudio5 qPCR system. A 1000× dye solution was diluted to a 2× concentration in 50 mM sodium phosphate buffer (pH 7.5). For each measurement, 10 μL of the diluted dye solution was combined with 10 μL of 0.8 mg/mL UGT in a 2× buffer solution into a 96-well qPCR plate. The plate was placed in the qPCR machine, where the temperature was initially stabilized at 25 °C for 2 min, and then gradually increased to 99 °C. Measurements were performed in triplicate, and data were analyzed as the mean from three independent experiments using Protein Thermal Shift™ software v1.x.

For the kinetic characterization, the activity of 0.1–0.6 mg/mL UGT was evaluated against 5–500 μM DON in optimum conditions for each enzyme (buffer at optimal pH and temperature). The experiments were performed under fixed UDP-Glc concentration (2 mM). The reaction was initiated by adding the acceptor substrate and quenched by adding 90 μL 100% MeOH after 15 min. *K*_m_ and *k*_cat_ values were determined by fitting the initial velocity data using the Michaelis–Menten model and the *drc* package in Rstudio (Rstudio version 2024.04.2 +764, RStudio, Inc., Vienna, Austria) [82].

### 4.7. LC–MS/MS and HPLC Analysis

In the GLY-it library screening assays, enzymatic reactions were evaluated using ultra-high-performance liquid chromatography coupled with DAD detection and tandem mass spectrometry (UHPLC–DAD–MS/MS). Analyses were conducted on an Agilent 1290 Infinity II UHPLC system, interfaced with a 1260 Infinity II DAD and an Ultivo Triple Quadrupole MS (Agilent Technologies, Santa Clara, CA, USA). Chromatographic separation was achieved on a Kinetex XB-C18 analytical column (1.7 μm, 100 Å, 100 × 2.1 mm, Phenomenex, Torrance, CA, USA). The mobile phases comprised Milli-Q water with 0.1% formic acid (phase A) and methanol with 0.1% formic acid (phase B). A gradient elution was performed at a flow rate of 0.4 mL/min, with the following program: 0–0.3 min, 5% B; 0.3–4.0 min, 5–35% B; 4.0–4.1 min, 35–98% B; 4.1–5.0 min, 98% B; 5.0–5.1 min, 98–5% B; 5.1–6.5 min, 5% B. The column was maintained at 30 °C, and analytes were detected at 245 nm. To confirm the presence of DON and DON-3-Glc, a multiple reaction monitoring (MRM) method was employed on the QqQ, which was operated in both Electrospray Positive Mode (ESI+) and Negative Mode (ESI−). Details regarding the MS source parameters are provided in Appendix A, and MRM transitions for the DON and DON-3-Glc are listed in Appendix A. Additionally, an unidentified peak (DON-X-G), eluting at 2.94 min, was detected and hypothesized to correspond to either DON-15-Glc or DON-7-Glc. Data were analyzed with MassHunter Quantitative Analysis software (Version 12.1, Agilent Technologies, Santa Clara, CA, USA).

For all in-house enzymatic assays, reaction mixtures were analyzed via reverse-phase high-performance liquid chromatography (HPLC) on an Ultimate 3000 Series system (Thermo Fisher Scientific, Waltham, MA, USA) equipped with a Kinetex C18 analytical column (2.6 μm, 100 Å, 100 × 4.6 mm, Phenomenex, Torrance, CA, USA). The mobile phases used were Milli-Q water containing 0.1% formic acid (phase A) and acetonitrile (phase B). The separation was achieved using a flow rate of 1 mL/min with the following gradient: 0–0.1 min, 2% B; 0.1–4.0 min, 2–70% B; 4.0–4.1 min, 70–100% B; 4.1–4.5 min, 100% B; 4.5–4.51 min, 2% B. The column was maintained at 40 °C, and analytes were detected at 240 nm. HPLC data acquisition and quantification were performed using Chromeleon software (Version 7.3.2, Thermo Fisher Scientific, Waltham, MA, USA).

### 4.8. Cloning of UGTs in Yeast

The genes encoding the UGTs were amplified from the GLY-it plasmids using PCR with specific primers (Integrated DNA Technologies, Coralville, IA, USA; Appendix A). HindIII and NotI restriction sites were incorporated at the 5′ and 3′ ends of the amplified genes, respectively. Expression plasmids were constructed by digesting the PCR products with HindIII and NotI (Thermo Fisher Scientific, Waltham, MA, USA) and ligating them into the pWS1921 backbone, which was also cleaved with these enzymes [32]. Gene sequences were confirmed by Sanger sequencing (Eurofins Genomics, Konstanz, Germany) using primers specific to the vector backbone. The UGT expression vectors, along with an empty pBP910 vector as a negative control and a plasmid expressing *Hv*UGT13248 as a positive control, were transformed into the mycotoxin-sensitive *S. cerevisiae* strain YZGA515 (*pdr5*Δ::*TRP1*, *pdr10*Δ::*hisG*, *pdr15*Δ::*loxP-KanMX-loxP*, *ayt1*Δ::URA3) [31]. Transformants were selected on a synthetic complete medium lacking leucine (SC–leu). Exponentially growing cultures were diluted to OD_600_ of 0.3 and 0.03 with fresh selective medium. Aliquots of 3 µL from these dilutions, in duplicate, were spotted onto yeast extract peptone dextrose (YPD) plates containing various concentrations of DON. The plates were incubated at 30 °C for 3 to 6 days.

### 4.9. Data Analysis and Visualization

Unless otherwise mentioned, all enzymatic assays were performed in duplicate, and results are presented as mean values ± standard deviation. Enzymatic conversion was quantified using calibration curves constructed from analytical standards when available. In cases where standards were not accessible, peak areas were used for relative quantification. Data processing, statistical analysis, and graphical visualization were conducted using RStudio (Rstudio version 2024.04.2 +764, RStudio, Inc.) [82].

## Figures and Tables

**Figure 1 toxins-17-00153-f001:**
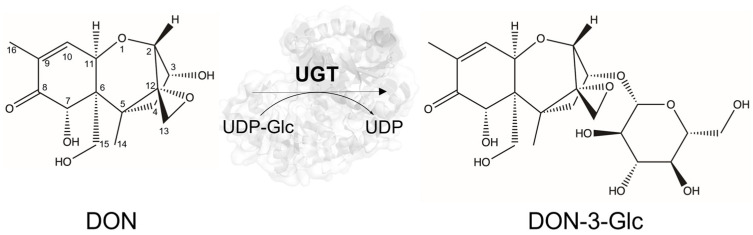
Chemical structure of deoxynivalenol (DON) and its primary detoxification product, deoxynivalenol-3-*O*-glucoside (DON-3-Glc), formed via enzymatic glycosylation catalyzed by UDP-glycosyltransferases (UGTs) with UDP-glucose (UDP-Glc) as the glucose donor. Although the main product identified in this study is DON-3-Glc, a second, yet unidentified product (DON-X-G) was also observed in some reactions.

**Figure 2 toxins-17-00153-f002:**
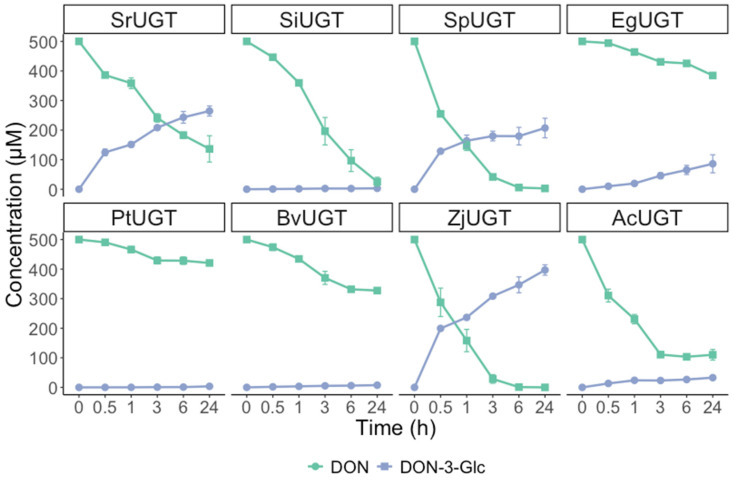
Time-course glycosylation reaction of DON (0–24 h) catalyzed by the recombinant UGTs identified in the GLY-it screening, resulting in the formation of DON-3-Glc. Error bars indicate the standard deviation from the mean of duplicate measurements. Each assay was conducted with 0.75 mg/mL of purified enzyme, 0.5 mM DON, and 2 mM UDP-Glc in a 50 mM sodium–phosphate buffer at pH 7.5. Reactions were quenched at specific time intervals and analyzed by reverse-phase HPLC. For enzymes with low DON-3-Glc production titers, a zoomed-in time-course plot is available in Appendix A.

**Figure 3 toxins-17-00153-f003:**
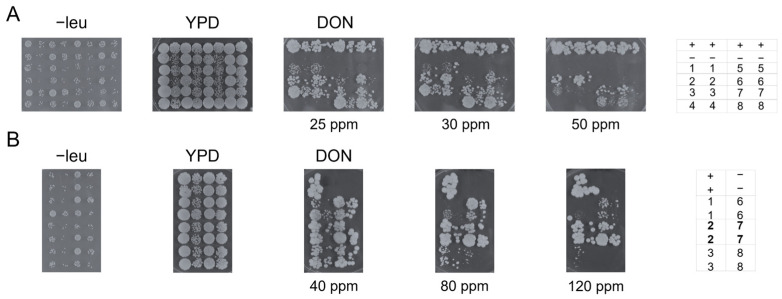
Spotting assays of yeast transformants expressing GLY-it UGTs on plates with varying DON concentrations in ppm (mg/L). (**A**) Comprehensive panel of UGTs tested at lower DON concentrations (25–50 ppm) after 3 days of incubation. (**B**) Close-up on UGTs that exhibit resistance to DON concentrations above 40 ppm after 6 days of incubation. Toxin-containing plates are prepared using YPD medium. Control plates include SC−leu, which supports the growth of only transformed yeast cells, and YPD, which permits the growth of strains without a plasmid. Two independent transformants of each construct were spotted in two dilutions (OD_600_ = 0.3 and 0.03, horizontally). A plate map for both experiments is provided on the right. UGTs conferring the highest resistance, almost comparable to the positive control, are indicated in bold in the plate map. Abbreviations: +: positive control (*Hv*UGT13248); − negative control (empty vector); 1: *Sr*UGT; 2: *Si*UGT; 3: *Sp*UGT; 4: *Eg*UGT; 5: *Pt*UGT; 6: *Bv*UGT; 7: *Zj*UGT; 8: *Ac*UGT.

**Table 1 toxins-17-00153-t001:** Novel UGTs identified in this study for activity on DON.

Enzyme	Organism	Subfamily	Group	Common Name	Accession (NCBI)
*Sr*UGT	*Stevia rebaudiana*	73	D	Stevia	SEQ ID NO 12 [54]
*Si*UGT	*Sesamum indicum*	73	E	Sesame	XP_011077288
*Sp*UGT	*Solanum pennellii*	73	C	Wild tomato	XP_015088999
*Eg*UGT	*Eucalyptus grandis*	76	H	Red grandis	XP_010033065
*Pt*UGT	*Populus trichocarpa*	73	D	Black cottonwood	KAI5604194
*Bv*UGT	*Beta vulgaris* subsp. *vulgaris*	73	D	Beet	KMT08362
*Zj*UGT	*Ziziphus jujuba* var. *spinosa*	71	E	Red date	WFR85808
*Ac*UGT	*Ananas comosus*	73	C	Pineapple	XP_020089421

**Table 2 toxins-17-00153-t002:** Biochemical and apparent kinetic properties of the top DON-3-Glc-producing UGTs identified in this study, along with the kinetic parameters of known DON UGTs from the literature. *n.a.*: *not available*.

Enzyme	Optimal pH	Optimal Temperature (°C)	T_M_ (°C)	*K*_m_ (mM)	*k*_cat_ (s^−1^)	*k*_cat_/*K*_m_ (M^−1^ s^−1^)	Reference
*Sr*UGT	8	45	59.4 ± 0.6	0.89 ± 0.18	0.80 ± 0.20	8.83 × 10^2^	This study
*Sp*UGT	8	32	54.3 ± 0.9	0.66 ± 0.14	0.40 ± 0.02	6.28 × 10^2^	This study
*Eg*UGT	8.5	30	46.9 ± 0.4	0.94 ± 0.19	0.04 ± 0.02	4.22 × 10^1^	This study
*Zj*UGT	8	39	48.1 ± 0.2	0.42 ± 0.14	0.93 ± 0.06	2.45 × 10^3^	This study
*Ac*UGT	8	45	63.0 ± 0.5	0.68 ± 0.12	0.02 ± 0.01	3.05 × 10^1^	This study
*Os*UGT79	*n.a.*	*n.a.*	*n.a.*	0.23 ± 0.06	0.57	2.48 × 10^3^	[34]
*Os*UGT79	*n.a.*	*n.a.*	*n.a.*	0.061	1.07 ± 0.04	1.75 × 10^4^	[40]
*Hv*UGT13248	*n.a.*	*n.a.*	*n.a.*	3.0 ± 0.6	0.78	2.60 × 10^2^	[42]

## Data Availability

The original contributions presented in this study are included in the article/Appendix A. Further inquiries can be directed to the corresponding author. The GLY-it sequence data do not contain novel protein sequences. The protein composition of the GLY-it screen is a third-party property and can be shared by permission of River Stone Biotech ApS (laurad@gly-it.com).

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
