# Peer review of "Plant-Derived UDP-Glycosyltransferases for Glycosylation-Mediated Detoxification of Deoxynivalenol: Enzyme Discovery, Characterization, and In Vivo Resistance Assessment"

_toxins, 2025, doi:10.3390/toxins17040153_

Round 1
Reviewer 1 Report
Comments and Suggestions for Authors
The authors screened a library of 380 plant UDP-GTs (called GLY-it library) for glycosylation activity against DON. Most were inactive, however, eight were observed to modify DON and were further characterized. Most produced DON-3-Glucoside, a known non-toxic biotransformation product of the toxin. Kinetics analysis indicated the best performing enzyme, ZjUGT, was on par with other known DON-UDP GTs from rice and barley. Interestingly, some of the UDP-GTs in the GLY-it library appeared to generate a novel DON glucoside that was simply referred to as DON-X-G. DON-X-G appeared to be non-toxic in S. cerevisiae assays.
Overall, the paper was potentially interesting, but there are a few critical issues that require addressing:
- Line 136. The author’s state that too low amounts of the compound were available. Is that because it was difficult to produce, or difficult to purify? The data for SiUGT in Figure 2 would seem to indicate that robust conversion of DON into DON-X-Glc is occurring, and that therefore it is a stability/purification issue with DON-X-Glc?
- Line 128-136. Why isn’t the HPLC data for DON-X-Glc and DON-15-Glc shown? Presumably, the authors would have enough of the DON-X-Glc for HPLC-MS analysis to know for sure that it is not DON-15-Glc or DON-3-Glc. That data should be shown. If not, their claims of DON-X-Glc remain too speculative to be included.
- In Figure S1, DON3G appears to elute after DON via UV and MS/MS analysis. Typically, DON3G elutes earlier than DON on a C18 column due to it being more polar. Can the authors provide an explanation for this?
- Section 5.2 and 5.3 are duplicated?
Author Response
The authors screened a library of 380 plant UDP-GTs (called GLY-it library) for glycosylation activity against DON. Most were inactive, however, eight were observed to modify DON and were further characterized. Most produced DON-3-Glucoside, a known non-toxic biotransformation product of the toxin. Kinetics analysis indicated the best performing enzyme, ZjUGT, was on par with other known DON-UDP GTs from rice and barley. Interestingly, some of the UDP-GTs in the GLY-it library appeared to generate a novel DON glucoside that was simply referred to as DON-X-G. DON-X-G appeared to be non-toxic in S. cerevisiae assays.
Overall, the paper was potentially interesting, but there are a few critical issues that require addressing:
- Line 136. The author’s state that too low amounts of the compound were available. Is that because it was difficult to produce, or difficult to purify? The data for SiUGT in Figure 2 would seem to indicate that robust conversion of DON into DON-X-Glc is occurring, and that therefore it is a stability/purification issue with DON-X-Glc?
We appreciate the reviewer’s insightful comment. The primary challenge was the overall low yield of DON-X-G, which remained insufficient for large-scale purification and high-resolution NMR characterization. While SiUGT and other enzymes did produce measurable amounts of DON-X-G, obtaining sufficient quantities for comprehensive structural analysis proved difficult. We have now explicitly stated this limitation in the manuscript (lines 145–149).
- Line 128-136. Why isn’t the HPLC data for DON-X-Glc and DON-15-Glc shown? Presumably, the authors would have enough of the DON-X-Glc for HPLC-MS analysis to know for sure that it is not DON-15-Glc or DON-3-Glc. That data should be shown. If not, their claims of DON-X-Glc remain too speculative to be included.
We thank the reviewer for this valuable suggestion. When we first screened the library and performed MS/MS analysis, we did not have an analytical standard for DON-15-Glc, which prevented a definitive identification. However, we did have an analytical standard for DON-3-Glc, which did not match DON-X-Glc (Figure S1 and lines 135-137). Despite this, we have now included a supplementary figure (Figure S7 and lines 140-142) displaying the HPLC chromatogram of the SiUGT reaction producing DON-X-Glc, alongside analytical standards for DON-15-Glc, DON-3-Glc, and DON.
- In Figure S1, DON3G appears to elute after DON via UV and MS/MS analysis. Typically, DON3G elutes earlier than DON on a C18 column due to it being more polar. Can the authors provide an explanation for this?
We greatly appreciate the reviewer’s comment. Indeed, DON-3-Glc is expected to elute earlier than DON on a C18 reversed-phase column due to its higher polarity. However, in our chromatographic separation, DON eluted before DON3G. We attribute this to the use of methanol as the organic phase (mobile phase B), which may have influenced retention times. To clarify this, we have added an explanatory note in the caption of Figure S1 and the main text (lines 105–107).
- Section 5.2 and 5.3 are duplicated?
We thank the reviewer for catching this redundancy. We have now corrected this issue by removing the duplicated text (lines 316-320).
Reviewer 2 Report
Comments and Suggestions for Authors
The experimental design of this manuscript is reasonable, and the research findings can broaden our understanding of UGT-mediated DON detoxification while demonstrating potential for agricultural applications. However, several revisions are required:
- Although repeated experiments were described in the methodology, the statistical analysis appears to be based on a limited number of replicates. It is recommended to employ more rigorous statistical approaches (e.g., ANOVA) to enhance the reliability of research outcomes.
- The four UGTs with lower activity generated uncharacterized products (DON-X-G). it is recommended to elucidate the specific structure of DON-X-G using NMR or more sensitive MS techniques.
- Discrepancies between in vitro activity and in vivo experimental results were observed for certain UGTs. So further optimization of experimental optimization or in-depth discussion regarding the potential causes of these inconsistencies between in vitro and in vivo findings is strongly recommended.
Author Response
The experimental design of this manuscript is reasonable, and the research findings can broaden our understanding of UGT-mediated DON detoxification while demonstrating potential for agricultural applications. However, several revisions are required:
- Although repeated experiments were described in the methodology, the statistical analysis appears to be based on a limited number of replicates. It is recommended to employ more rigorous statistical approaches (e.g., ANOVA) to enhance the reliability of research outcomes.
We appreciate the reviewer’s suggestion and fully agree that rigorous statistical analysis is essential. However, our statistical approach follows standard practices in enzymology and enzyme characterization, where duplicate or triplicate measurements are common due to sample limitations. We report the mean ± standard deviation to represent data variability. In addition, to our knowledge, and upon reviewing the literature after the reviewer’s comment, ANOVA is not appropriate to use here, since enzyme kinetic data are not suitable for linear analysis, often exhibit non-independent error structures, and therefore require non-linear regression for accurate interpretation (e.g., Michaelis-Menten kinetics). To ensure clarity, we have now explicitly stated our statistical methodology in the Materials and Methods section (Section 4.9, lines 425-430).
- The four UGTs with lower activity generated uncharacterized products (DON-X-G). it is recommended to elucidate the specific structure of DON-X-G using NMR or more sensitive MS techniques.
We appreciate the reviewer’s suggestion. While we acknowledge the importance of fully characterizing DON-X-Glc, obtaining sufficient material for purification and subsequent high-resolution NMR analysis was not feasible within this study. We have now explicitly discussed this limitation in the text, emphasizing that future work will focus on upscaling enzymatic reactions and optimizing purification strategies to enable structural elucidation (lines 145–149).
- Discrepancies between in vitro activity and in vivo experimental results were observed for certain UGTs. So further optimization of experimental optimization or in-depth discussion regarding the potential causes of these inconsistencies between in vitro and in vivo findings is strongly recommended.
We fully agree with the reviewer that understanding these discrepancies is crucial. We have now expanded our discussion on the possible reasons for the differences observed between in vitro enzymatic activity and in vivo yeast resistance assays, including factors such as heterologous expression in yeast, UGT promiscuity, enzyme localization, and cofactor requirements (lines 223–259).
Reviewer 3 Report
Comments and Suggestions for Authors
In this manuscript, the authors screened a library of 380 recombinant plant UDP-glycosyltransferases (UGTs), identified and characterized eight novel enzymes capable of glycosylating deoxynivalenol (DON), and evaluated the in vivo resistance conferred by these UGTs when expressed in a DON-sensitive yeast strain. There are still some issues to be more specific analysis and explanation. Please check the attachment for details.

The overall quality of the English language in this paper is relatively high. It demonstrates good normativity and professionalism in terms of vocabulary, grammar, discourse structure, and academic expression, and can effectively convey the research content. However, there is still room for improvement in some details.
Author Response
In this manuscript, the authors screened a library of 380 recombinant plant UDP-glycosyltransferases (UGTs), identified and characterized eight novel enzymes capable of glycosylating deoxynivalenol (DON), and evaluated the in vivo resistance conferred by these UGTs when expressed in a DON-sensitive yeast strain. There are still some issues to be more specific analysis and explanation.
- In section 2.1, there's a lack of specific basis for screening enzymes in the GLY-it library: The manuscript only mentions screening the library but doesn't provide criteria for selecting the 380 UGTs. It's necessary to supplement the screening basis.
We appreciate the reviewer’s suggestion. We have explicitly stated (lines 100–102) that the GLY-it library was designed to include UGTs that are well-expressed in E. coli, as well as substrate-promiscuous, and phylogenetically diverse, ensuring broad coverage of plant UDP-glycosyltransferases with potential detoxification activity on DON.
- Section 2.2 lacks blank experiments: There are no blank experiments to rule out factors such as DON's spontaneous degradation or buffer interference. Negative controls should be added, and a control dataset in Figure 2 or Figure S5 is needed to verify enzyme-dependent DON conversion.
We thank the reviewer for this observation. We have now explicitly stated in the text that our experiments included appropriate controls, confirming that reaction conditions had no impact on DON stability and that no spontaneous degradation of DON was observed in the absence of enzymes (lines 129–131). Additionally, we have provided a representative HPLC chromatogram illustrating DON conversion in the presence of UGTs, as well as the absence of DON conversion in the absence of enzymes (Figure S6).
- Regarding Figure 3: In Figure 3, only images show yeast growth. Quantitative data like OD values or colony counts should be added. Growth curves or bar graphs should be included to quantify yeast resistance to DON. At the same time, it is better to make pointer annotations to facilitate readers' better understanding.
We appreciate the reviewer’s recommendation. While we agree that including growth curves would provide a more quantitative assessment, it is unfortunately not feasible to generate these data at this stage. However, we have now explicitly described the yeast selection, growth, and dilution procedures in the caption of Figure 3 and Section 4.8 (lines 417–421). To improve clarity, we have also highlighted in bold the yeast transformants overexpressing the most resistant UGTs in the plate map in Figure 3.
- Why were different receptors selected for the expression and verification process, especially Escherichia coli, which is a model organism of prokaryotes, and Saccharomyces cerevisiae, a model organism of eukaryotes?
We appreciate the reviewer’s question and have now explicitly stated (lines 199–202) the rationale behind this choice. E. coli was selected for enzyme expression due to its efficiency in producing recombinant proteins, while S. cerevisiae was used for functional validation as it provides a relevant eukaryotic system to assess DON detoxification, in line with previous studies evaluating enzymes in a cellular context.
- Regarding the supplementary materials, please verify them against the experimental data. For example, for Figure S2 in the supplementary materials: In Figure S2, the y-axis “peak area” scale is too large and error bars are missing. The scale should be adjusted and error bars added for better data clarityï¼›For Figure S4 in the supplementary materials: The SDS-PAGE lacks molecular weight markers. Please label the molecular weights on the lanes.
We appreciate the reviewer’s concerns and have carefully reviewed Figure S2. The scale was intentionally adjusted to accurately reflect the experimental data from the initial library screening, particularly with regard to DON consumption, to ensure that the presentation was not misleading. Additionally, we have now explicitly noted in the figure caption and in the main text (lines 307–308) why error bars were not included in this dataset from the initial library screening.
Regarding Figure S4, we have now added molecular weight markers to the SDS-PAGE figure.
Other aspects that need to be carefully confirmed include:
- The experimental methods in sections 5.2 and 5.3 are repetitive: These two sections describe overlapping experimental steps.
We thank the reviewer for pointing this out. We have now corrected this.
- Please check the section numbering: The section numbering is inconsistent.
We appreciate the reviewer’s careful review. The section numbering inconsistencies have now been corrected.
- Literature citations: Literature citations in the research status are scattered.
We thank the reviewer for pointing this out. We have revised and structured the literature citations more clearly.
- The entire text needs to be refined. In particular, the Introduction should clearly state the limitations of current research and the study's breakthroughs, and the Conclusion also requires refinement.
We thank the reviewer for the valuable suggestion. The text has been refined, with particular attention to the Introduction and Conclusion sections. The Introduction now explicitly highlights the limitations of current research and the breakthroughs of our study (lines 75-78, lines 84-85 and lines 87-89), while the Conclusion has been revised to better emphasize our findings (lines 282-284, and lines 287-290).
Round 2
Reviewer 1 Report
Comments and Suggestions for Authors
The updated version of the manuscript is acceptable for publication.
Reviewer 3 Report
Comments and Suggestions for Authors
The authors have made the revisions as required, and there are no other issues.
Comments on the Quality of English LanguageThe authors have made the revisions as required, and there are no other issues.